# Optimization of Fermentation Process of Wheat Germ Protein by *Aspergillus niger* and Analysis of Antioxidant Activity of Peptide

Yingying Liu [1], Yu Zhou [1], Chaohong Zhu [1], Yanglin Meng [1], Jingjing Wang [1], Xinyang Chen [1], Yinchen Hou [2], Aimei Liao [1], Long Pan [1,*] and Jihong Huang [1,2,3,4,*]

1. College of Biological Engineering, Henan University of Technology, Zhengzhou 450001, China; 2021920157@stu.haut.edu.cn (Y.L.); 221060100413@stu.haut.edu.cn (Y.Z.); 202012@stu.haut.edu.cn (C.Z.); 2962286710@stu.haut.edu.cn (Y.M.); 2021930602@stu.haut.edu.cn (J.W.); 2021930593@stu.haut.edu.cn (X.C.); aimeiliao@haut.edu.cn (A.L.)
2. Food Laboratory of Zhongyuan, Luohe 462000, China; 80891@hnuahe.edu.cn
3. State Key Laboratory of Crop Stress Adaptation and Improvement, College of Agriculture, Henan University, Kaifeng 475004, China
4. School of Food and Pharmacy, Xuchang University, Xuchang 461000, China
* Correspondence: panlong15890988310@126.com (L.P.); huangjihong@haut.edu.cn (J.H.)

**Abstract:** Utilizing wheat embryos as the raw material and employing *Aspergillus niger* as the fermentation strain, wheat embryo polypeptides were produced through microbial liquid fermentation. The protein concentration post-fermentation served as the response variable, and the preparation process underwent optimization through single-factor testing and a response surface methodology, followed by the assessment of antioxidant activity. The findings revealed that the optimal conditions for wheat embryo peptide preparation via *Aspergillus niger* fermentation included a fermentation duration of 24 h, an inoculum volume of 4%, an initial pH of 7, and a protein concentration of 21.47 mg/mL. Peptides with different molecular weights were then prepared by dead-end filtration. The results showed that F6 (<3 kDa) had strong scavenging ability against DPPH, ABTS, and ·OH radicals, which provided a basis for the preparation of antioxidant peptides in wheat germ and related research.

**Keywords:** wheat germ protein; protease; biopeptides; fermentation; *Aspergillus niger*

## 1. Introduction

Wheat germ is a by-product of wheat processing with a protein content exceeding 30%, making it a potential plant protein resource [1–3]. Among these proteins, albumin accounted for 34.5%, globulin accounted for 15.6%, gliadin accounted for 4.6%, glutenin accounted for 10.6%, and insoluble residues accounted for 34.7% [4]. Wheat germ protein is versatile, serving as a natural high-quality food protein fortification agent that supplements protein in food and reinforces amino acids [5]. Wheat germ protein exhibits comparable essential amino acid ratios to those in egg and milk proteins, closely aligning with the modeled values published by the FAO/WHO, positioning it as a plant protein source with developmental potential [6].

Peptides and protein hydrolysates have emerged as contemporary sources of natural antioxidants [7,8]. Besides wheat germ proteins themselves, the production of peptides through enzymatic and biofermentation methods, along with subsequent investigations, has been extensively reported. Enzymolysis involves the degradation of proteins by either endogenous or exogenous proteases [9]. Wheat germ proteins encompass numerous active amino acid sequences. Under the influence of protein hydrolases, specific peptide bonds undergo hydrolysis, resulting in the generation of abundant peptide compounds. This formation contributes to enhancing certain functional properties and manifesting specific biological activities. Presently, researchers have isolated active peptides exhibiting anti-oxidative-stress [10] and antioxidant peptides [11,12] and investigated the mechanism

of action underlying antioxidant peptides [13,14]. Karami employed a response surface methodology to optimize the antioxidant capacity during the enzymatic hydrolysis of wheat germ protein. Subsequently, the fraction with higher antioxidant capacity was sequenced [15].

Fermentation is an effective technique for producing or extracting antioxidant-active compounds from natural substances. Compared with enzymatic hydrolysis, microbial production of enzymes can be used for enzymatic hydrolysis, which reduces the production cost. Fermented wheat germ extracts have been shown to possess biological activities such as antioxidant [16,17], anti-inflammatory [18], and anticancer [19], suggesting the value of wheat germ for further development and utilization. Fermentation of wheat germ ethanol extract revealed that alterations in bioactive substances during the process enhanced scavenging activity against transition-metal-induced oxidative stress, exhibiting stronger antioxidant effects [20]. Fermented wheat germ globulin has also been reported to have anti-aging effects and to intervene in abnormal glucose-lipid metabolism and chronic hypertension [21,22]. Starzyńska [23] employed an edible strain of *Rhizobium* to ferment wheat germ cake. The fermentation increased peptide content, antioxidant activity, and the levels of essential amino acids such as lysine and leucine. Liu [17] employed the fermentation of defatted wheat germ by *Bacillus subtilis*. The study revealed that peptides exerted a more significant impact on antioxidant activity than phenolic compounds in the early stages of fermentation, with both exhibiting a synergistic effect. Fermentation of wheat germ by *Bacillus subtilis* has been reported to enhance the release of peptides, phenolic compounds, and other bioactive substances [24]. Wu [25] utilized *Lactobacillus plantarum* to ferment wheat germ, optimizing the fermentation time, temperature, initial pH, and feed-to-liquid ratio to increase the soluble protein content of the fermentation products.

Currently, peptide separation primarily involves membrane separation technology and chromatographic separation technology, with the former being most widely employed, especially through ultrafiltration. Centrifugal ultrafiltration filters with different molecular weight cutoff membranes have been used to separate peptides from pinto bean protein isolate. The final result was a peptide with a molecular weight < 3 kDa, with a higher antioxidant activity, which may be due to the fact that small-molecule peptides react more easily with lipid free radicals and thus reduce lipid peroxidation [26].

While significant progress has been achieved in the hydrolysis of various raw materials and bioactive peptides through microbial fermentation, particularly in the development of soybean peptides and marine fish protein peptides, fewer studies have explored the preparation of highly active wheat germ protein peptides using microbial fermentation. This study aims to produce wheat germ peptides through microbial hydrolysis of proteins extracted from defatted wheat germ. This process not only aims to reduce the production costs associated with wheat germ peptides but also seeks to offer insights into peptide preparation methodologies.

## 2. Materials and Methods

### 2.1. Materials and Equipment

Defatted wheat germ powder was purchased from Henan Kunhua Biotechnology Co, Ltd. (Zhengzhou, China), while 1,1-diphenyl-2-picryl-hydrazyl radical (DPPH), 2,2′-Azinobis-(3-ethylbenzthiazoline-6-sulphonate) (ABTS), high-temperature alpha amylases, glucosidases, and other chemical reagents were obtained from Shanghai McLean Biochemical Technology Co., Ltd. (Shanghai, China). Bovine serum protein standards were obtained from Nanjing Jiancheng Biological Research Institute. $CuSO_4$, Seignette salt, NaOH, vitamin C, $FeSO_4$, hydrogen peroxide, salicylic acid, and so on are analytically pure reagents. *Bacillus subtilis*, *Lactobacillus acidophilus*, *Lactobacillus plantarum*, *Bacillus coagulans*, *Bacillus natto*, *Aspergillus Niger,* and *Aspergillus oryzae* were obtained from the laboratory collection (Henan Industrial Microbial Strain Collection Center of Henan Province). Tangential flow filter membrane packages were obtained from Sartorius AG, Ltd. (Göttingen, Germany), and ultrafiltration tubes from Sigma Aldrich Trading Co., Ltd. (Shanghai, China).

Constant-temperature culture shaker was obtained from Shanghai Zhicu Co., Ltd. (Shanghai, China); ultra-clean table from Zhejiang Purification Equipment Co., Ltd. (Shaoxing, China); high-speed centrifuge from Xiangyi Instrument Co., Ltd. (Changsha, China) and Eppendorf Company (Hamburg, Germany); freeze dryer from Ningbo Xinzhi Biotechnology Co., Ltd. (Ningbo, China); and multifunctional microplate reader from BioTek Instruments Inc. (Winooski, VT, USA).

### 2.2. Preparation of Wheat Germ Protein Fermentation Medium

Add wheat germ powder and 0.9% KCl solution in a liquid-to-solid ratio of 1:10 (g/mL). After thorough stirring, seal with gauze and oscillate in a shaker for 1.5 h (150 rpm, 30 °C). Adjust the pH to 6.0–7.0 after oscillation. Perform liquefaction treatment by adding high-temperature α-amylase at 95 °C for 2 h with continuous stirring. Cool the conical flask after oscillation, adjust the pH to 4.6 ± 0.5 with citric acid, and heat in a 40 °C water bath after adding glucoamylase. After completion, dispense into 250 mL conical flasks and sterilize in an 85 °C water bath for 30 min.

### 2.3. Activation of the Strain and Preparation of Seed Solution

For *Bacillus subtilis* and *Bacillus natto*, the activated bacterial solution was streaked onto LB solid agar plates. Single colonies were selected and then inoculated into liquid LB medium. Subsequently, the cultures were maintained at 37 °C for 24 h to establish the seed solution. For *Lactobacillus acidophilus*, *Lactobacillus plantarum*, and *Bacillus coagulans*, the activated bacterial solution was streaked onto MRS solid agar plates. Single colonies were isolated and subsequently inoculated into MRS medium. The cultures were then incubated at 37 °C for 24 h to establish the seed solution. For *Aspergillus niger* and *Aspergillus oryzae*, the preserved spore solution was inoculated onto PDA solid agar plates. Following an incubation period of 5 days at 28 °C, the spores were washed with sterile water, then inoculated into liquid potato medium and further incubated at 28 °C for 24 h to establish the seed solution.

### 2.4. Screening of Biomodified Strains

The reducing power served as the index for selecting optimal biological fermentation strains. The prepared medium was inoculated with *Bacillus subtilis*, *Lactobacillus acidophilus*, *Lactobacillus plantarum*, *Bacillus coagulans*, *Bacillus natto*, *Aspergillus Niger*, and *Aspergillus oryzae* for the fermentation of wheat germ protein. The inoculum was 3%, the fermentation temperature was the same as that of the seed liquid, and the fermentation time was 24 h. After fermentation, the samples were centrifuged at 8000 rpm, at 4 °C, for 10 min. The optical density at 540 nm ($OD_{540}$) of the supernatant obtained after fermentation for each bacterial strain was measured to assess its reducing capacity. At this juncture, the sample with the lowest protein concentration measured 4.28 mg/mL. Consequently, all samples were uniformly diluted to this concentration. Additionally, a solution of vitamin C at a concentration of 4.28 mg/mL (24.3016 mM) was prepared to serve as the positive control.

Previous research indicates a correlation between reducing capacity and antioxidant activity, with stronger reducing capacity associated with increased antioxidant activity [27,28]. Compounds possessing robust reducing power, such as proteins and other antioxidants, can convert $Fe^{3+}$ to $Fe^{2+}$, and the extent of reduction is evaluated through a color reaction. A higher absorbance post-reaction indicates a greater reducing power of the substance. Add 1 mL of the test sample and 1 mL of 1% potassium ferrocyanide to 2.5 mL of pH 6.6 phosphate buffer solution, maintaining the mixture at 50 °C for 20 min. Subsequently, introduce 1 mL of 10% trichloroacetic acid, followed by centrifugation at 3000 r/m for 10 min. Retrieve the upper clear liquid, add 2.5 mL of distilled water, and introduce 0.5 mL of 0.1% FeCl₃. Ensure thorough mixing and allow it to stand for 10 min. Record the absorbance value at 700 nm, with all measurements representing the average of three trials. Vitamin C served as the positive control, with a concentration equivalent to that of the protein samples utilized for absorbance determination.

*2.5. Single-Factor Experiments for Optimization of Wheat Germ Protein Fermentation Conditions*

Employing a 4% *Aspergillus niger* inoculum, a 24 h fermentation time, and a temperature of 30 °C as the foundational parameters, we investigated the impact of varying Aspergillus niger inoculum volume (2%, 3%, 4%, 5%, 6%), initial pH (5, 6, 7, 8, 9), and fermentation time (20 h, 24 h, 28 h, 32 h, 36 h) on the supernatant protein concentration following wheat germ protein fermentation, manipulating one variable at a time.

*2.6. Response Surface Method for Optimization of Wheat Germ Protein Fermentation Conditions*

Employing the one-way test, we identified fermentation time (A), inoculum volume (B), and initial pH (C) as the three influential factors. The supernatant protein concentration after fermentation (Y) served as the response value for the design of a three-factor, three-level response surface test, and the factors and levels of this test are detailed in Table 1.

**Table 1.** Factors and levels of Box–Behnken experiments for fermentation conditions optimization of wheat germ protein.

| Level | A Fermentation Time/(h) | B Inoculation Volume/% | C Initial pH |
|:---:|:---:|:---:|:---:|
| −1 | 20 | 3 | 6 |
| 0 | 24 | 4 | 7 |
| 1 | 28 | 5 | 8 |

*2.7. Preparation of Peptides of Different Molecular Weights*

The solution post-fermentation of *Aspergillus niger* underwent refinement, and molecular peptides of varying molecular weight sizes were acquired by sequentially traversing filter membranes with molecular weights of 100 kDa, 50 kDa, 30 kDa, 10 kDa, and 3 kDa. This process yielded six fractions: F1 (>100 kDa), F2 (50–100 kDa), F3 (30–50 kDa), F4 (10–30 kDa), F5 (3–10 kDa), and F6 (<3 kDa). We investigated the antioxidant activities of peptides with varying molecular weights, assessing indices such as DPPH radical scavenging capacity, ·OH radical scavenging capacity, and ABTS radical scavenging capacity.

*2.8. Analysis of Antioxidant Effects of Peptides with Different Molecular Weights*

2.8.1. DPPH Free Radical Scavenging Capacity

The method outlined by Wang [29] and Yang [30] underwent slight modifications. Specifically, 5.0 mg of DPPH was weighed and dissolved in an appropriate quantity of absolute ethanol, fully dissolving through ultrasound in darkness. Subsequently, the volume was adjusted to 100 mL with absolute ethanol, and the solution was used for immediate preparation, as follows: (1) Add the sample solution and DPPH solution into the centrifuge tube in a 1:2 ratio, with a total volume of 750 μL. Incubate for 30 min shielded from light, then measure the absorbance value ($A_1$) at 517 nm using an enzyme meter. (2) Substitute the DPPH solution with anhydrous ethanol and measure the absorbance value ($A_2$) under identical conditions. (3) Substitute the protein sample solution with deionized water and measure the absorbance value ($A_0$) under identical conditions. (4) Prepare a vitamin C solution with the equivalent concentration as the protein solution to serve as a positive control, then subject it to the same conditions for reaction with the DPPH solution, measuring the absorbance at 517 nm using an enzyme meter. The sample with the lowest protein concentration among those of varying molecular weights was determined to be 4.38 mg/mL. Consequently, all samples were uniformly diluted to this concentration. Furthermore, a solution of vitamin C at a concentration of 4.38 mg/mL (24.8694 mM) was prepared to serve as the positive control.

$$\text{DPPH radical scavenging} = \left[1 - \frac{(A_1 - A_2)}{A_0}\right] \times 100\%$$

### 2.8.2. ·OH Free Radical Scavenging Capacity

The steps were as follows: (1) In a 2 mL centrifuge tube, combine 6 mM $FeSO_4$ solution, 3 mM hydrogen peroxide solution, and the sample solution in a 1:1:1 ratio, yielding a total volume of 1.5 mL. After thorough mixing, allow to stand for 10 min, then introduce 0.5 mL of 6 mM salicylic acid solution. Shake the mixture again and incubate for 30 min. Subsequently, measure the absorbance ($A_1$) at 510 nm using an enzyme meter. (2) Substitute the salicylic acid with deionized water and measure the absorbance ($A_2$) of the blank group under identical conditions. (3) Substitute the sample solution with deionized water and measure the absorbance ($A_0$) of the control group under identical conditions. (4) A vitamin C solution of the same concentration as the protein was prepared and reacted under the same conditions, and the absorbance was measured at 510 nm. The sample with the lowest protein concentration among those of varying molecular weights was determined to be 4.38 mg/mL. Consequently, all samples were uniformly diluted to this concentration. Furthermore, a solution of vitamin C at a concentration of 4.38 mg/mL (24.8694 mM) was prepared to serve as the positive control.

$$\text{Hydroxyl radical scavenging} = \left[ 1 - \frac{(A_1 - A_2)}{A_0} \right] \times 100\%$$

### 2.8.3. ABTS Free Radical Scavenging Capacity

Refer to the method of Sung-Sook Chun [31] and Kriengsak [32]. First, 5 mL of 7 mmol/L ABTS solution was mixed with 88 μL of 140 mmol/L potassium persulfate. Then, the mixture was left to stand overnight at room temperature in the absence of light and subsequently utilized as the ABTS radical stock solution. The solution was then diluted to achieve an absorbance of $A_{734} = 0.700 \pm 0.002$ before being employed as the working solution. During the determination process, 4 mL of ABTS working solution was thoroughly mixed with 1 mL of the sample solution and left at 25 °C for 6 min. The absorbance value $A_2$ was measured at 734 nm. For the blank group, 4 mL of DPPH solution and deionized water as the protein sample solvent were mixed evenly and reacted at 25 °C for 6 min. The absorbance was measured at 734 nm and recorded as $A_1$. The sample with the lowest protein concentration among those of varying molecular weights was determined to be 4.38 mg/mL. Consequently, all samples were uniformly diluted to this concentration. Furthermore, a solution of vitamin C at a concentration of 4.38 mg/mL (24.8694 mM) was prepared to serve as the positive control.

$$\text{ABTS radical scavenging activity} = \frac{(A_1 - A_2)}{A_1} \times 100\%$$

### 2.9. Statistical Analysis

All tests and assays in this study were repeated three times, and values are expressed as mean ± standard deviation. Data were analyzed for significance using IBM SPSS version 25.0 statistical software, and values were considered significantly different at $p < 0.05$. Data were plotted using Origin 2023b and GraphPad Prism 8 software. Response surface experimental design and data processing were carried out using Design Expert V13 software.

## 3. Results

### 3.1. Comparison of the Reducing Power of Different Strains of Fermentation

The determination of reducing power was based on the amount of Prussian blue $Fe_4[Fe(CN)_6]$ produced. The red blood salt $K_3Fe(CN)_6$ was reduced to yellow blood salt by antioxidants, and then $Fe^{2+}$ was used to form Prussian blue. The absorbance value at 700 nm reflected the amount of Prussian blue produced. The higher the absorbance value, the stronger the reducing power (antioxidant capacity) of the sample, indicating a better antioxidant effect. As shown in Figure 1, using *Bacillus subtilis*, *Lactobacillus acidophilus*,

*Lactobacillus plantarum, Bacillus natto, Bacillus coagulans, Aspergillus cereus,* and *Aspergillus Niger* to ferment wheat germ protein, the absorbance of *Aspergillus Niger* fermentation supernatant at 700 nm was 1.08, which was second only to the absorbance of VC. Therefore, the fermentation supernatant of *Aspergillus Niger* had high reducing power.

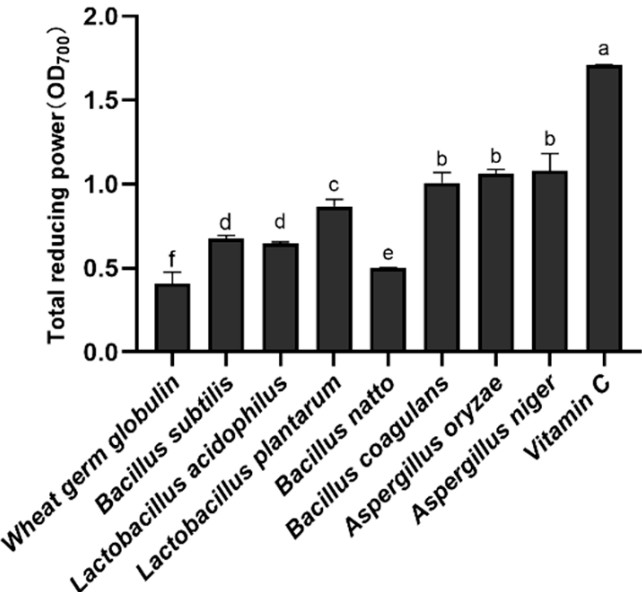

**Figure 1.** Reducing power of fermented wheat germ protein supernatants of different strains (Different corresponding letters indicate significant differences at $p < 0.05$ by Duncan's test).

*3.2. Single-Factor Experiments for Optimization of Wheat Germ Protein Fermentation Conditions*

Figure 2a illustrates that the protein concentration initially rises and subsequently declines with increasing fermentation time, reaching a peak concentration of 18 mg/mL at 24 h. This phenomenon can be attributed to the relatively stable growth rate of *Aspergillus niger* and the consequent increase in protein concentration during the initial fermentation stages. Prolonged fermentation durations lead to increased production of metabolic waste, compromised enzyme synthesis, and subsequent reduction in protein concentration. Figure 2b demonstrates a pattern wherein the protein concentration initially rises and subsequently declines with increasing inoculation amount. This phenomenon may stem from the inhibitory effect of excessive inoculation amounts, coupled with limited nutrient availability, which suppresses the growth of *Aspergillus niger* and consequently reduces polypeptide content and protein concentration. Figure 2c illustrates a trend where the protein concentration initially rises and subsequently declines with increasing pH value. This behavior could be attributed to the existence of an optimal pH range within the fermentation process of *Aspergillus niger*, where both excessively high and low pH levels inhibit its growth.

*3.3. Response Surface Method for Optimization of Wheat Germ Protein Fermentation Conditions*

According to the single-factor experiment, the optimum fermentation time was 24 h, the optimum inoculation amount was 4%, and the optimum initial pH was 7. According to the results of single-factor experiment, fermentation time, inoculation amount, and initial pH were selected as the main factors to investigate, and protein concentration was taken as the response value to conduct the response surface test. The experimental design scheme of the response surface is shown in Table 1. The experimental design and results are shown in Table 2.

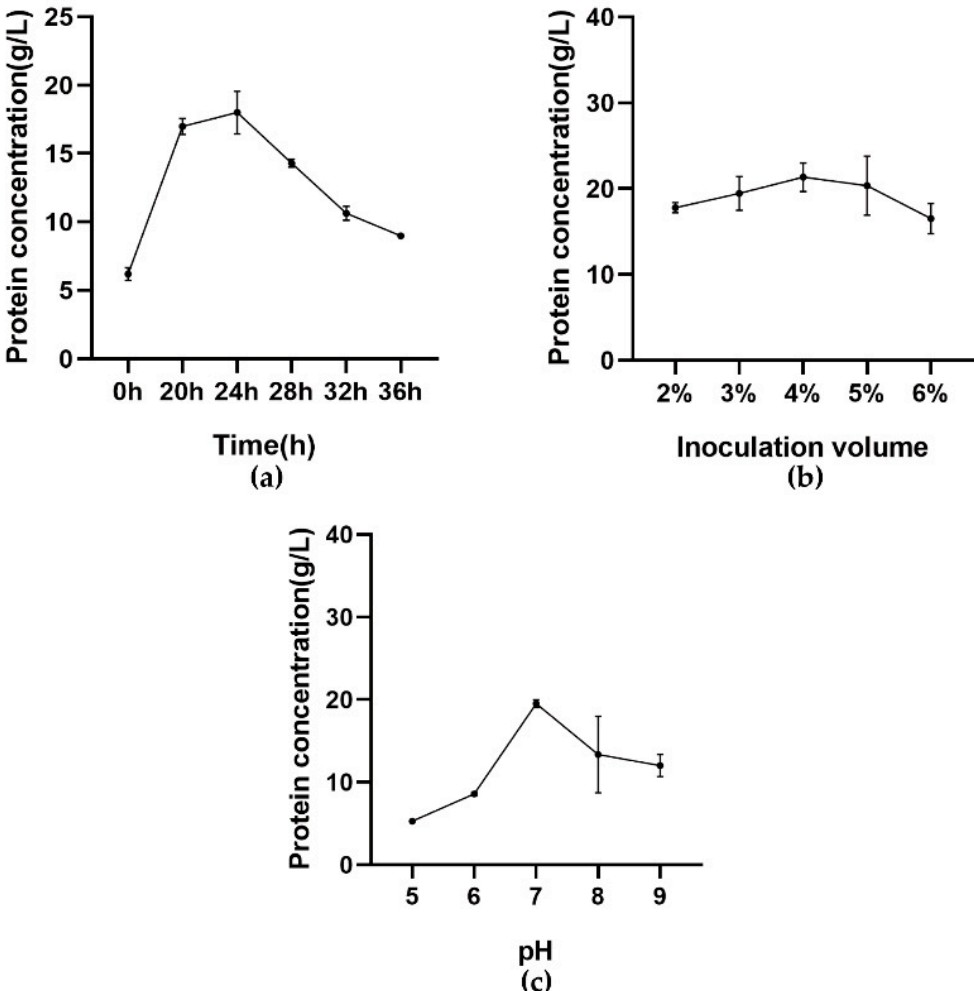

**Figure 2.** Effect of fermentation time (**a**), inoculum volume (**b**), and pH (**c**) on protein concentration.

**Table 2.** Response surface experimental design and results of wheat germ protein fermentation.

| Level | A Fermentation Time/(h) | B Inoculation Volume/% | C Initial pH | Protein Concentration (mg/mL) |
|---|---|---|---|---|
| 1 | 20 | 3 | 7 | 17.4849 |
| 2 | 28 | 3 | 7 | 15.9335 |
| 3 | 20 | 5 | 7 | 19.009 |
| 4 | 28 | 5 | 7 | 17.7434 |
| 5 | 20 | 4 | 6 | 10.15 |
| 6 | 28 | 4 | 6 | 11.5516 |
| 7 | 20 | 4 | 8 | 15.3892 |
| 8 | 28 | 4 | 8 | 12.9941 |
| 9 | 24 | 3 | 6 | 13.3071 |
| 10 | 24 | 5 | 6 | 11.4019 |
| 11 | 24 | 3 | 8 | 14.6271 |
| 12 | 24 | 5 | 8 | 17.2671 |
| 13 | 24 | 4 | 7 | 21.2544 |
| 14 | 24 | 4 | 7 | 20.7509 |
| 15 | 24 | 4 | 7 | 21.7715 |
| 16 | 24 | 4 | 7 | 21.023 |
| 17 | 24 | 4 | 7 | 21.7443 |

Design-Expert 13 was employed to analyze the data presented in Table 2, conduct individual significance analyses, and formulate a model. The results are documented in Table 3, where the regression model ($p < 0.0001$) attests to its considerable significance.

In the first term, the impact of factors A, B, and C on protein concentration proved to be significant. The missing item, with a *p*-value of 0.2362, did not achieve significance. The model exhibits a coefficient of determination ($R^2$) of 0.9914, a corrected coefficient of determination ($R^2$) of 0.9804, and a coefficient of variation (C.V. %) of 3.28%, which indicates that 99.14% of the variation in the response value originates from the selected variables, and this value closely approaches 100%, signifying that the optimized regression model better captures the real relationship between the operational conditions and protein concentration, making it suitable for determining optimal fermentation process parameters for wheat germ protein. The optimized regression model can be used to determine the optimal fermentation process parameters of wheat germ protein. Following the fitting of factors through quadratic polynomial regression, equations for the three factors—fermentation time, inoculum volume, and initial pH—were derived as follows: protein concentration $Y = 21.31 - 0.4763A + 0.5086B + 1.73C + 0.0714AB - 0.9492AC + 1.14BC - 2.7A^2 - 1.07B^2 - 6.09C^2$. A more pronounced slope in the response surface plot indicates a stronger influence of the interaction between the two factors on the index. Observing Figure 3 reveals that the interaction between fermentation time and initial pH (AC) and inoculum volume and initial pH (BC) significantly impacted protein concentration, aligning with the outcomes of the ANOVA. The transition from blue to red in the figure signifies a progression in protein concentration from low to high, with the speed of color change correlating to the steepness of the gradient. Figure 3b reveals that the interaction of AC has a more pronounced influence on protein concentration. These findings indicate that the inoculum volume and initial pH are primary factors influencing the increase in protein concentration, followed by fermentation time and initial pH. Design-Expert analyzed the results of the response surface tests, revealing the optimal process conditions for wheat germ protein fermentation: fermentation time of 23.534 h, inoculum volume of 4.331%, and initial pH of 7.182. Under these conditions, the theoretical protein concentration after fermentation was 21.58 mg/mL. The optimal process conditions were adjusted to a fermentation time of 24 h, an inoculum volume of 4%, and an initial pH of 7. Three validation tests were conducted under these conditions, and the protein concentration after fermentation was 21.47 mg/mL. The difference between the experimental and predicted values was small, and the model was reliable.

**Table 3.** Variance analysis of the regression model.

| Source | Sum of Squares | df | Mean Square | F-Value | *p*-Value |
|---|---|---|---|---|---|
| Model | 241.58 | 9 | 26.84 | 89.84 | <0.0001 |
| A-Time | 1.81 | 1 | 1.81 | 6.07 | 0.0432 |
| B-Inoculum Volume | 2.07 | 1 | 2.07 | 6.93 | 0.0338 |
| C-pH | 24.04 | 1 | 24.04 | 80.45 | <0.0001 |
| AB | 0.0204 | 1 | 0.0204 | 0.0683 | 0.8013 |
| AC | 3.6 | 1 | 3.6 | 12.06 | 0.0104 |
| BC | 5.16 | 1 | 5.16 | 17.29 | 0.0043 |
| $A^2$ | 30.65 | 1 | 30.65 | 102.57 | <0.0001 |
| $B^2$ | 4.8 | 1 | 4.8 | 16.08 | 0.0051 |
| $C^2$ | 156.15 | 1 | 156.15 | 522.62 | <0.0001 |
| Residual | 2.09 | 7 | 0.2988 | | |
| Lack of Fit | 1.29 | 3 | 0.4306 | 2.15 | 0.2362 |
| Pure Error | 0.7996 | 4 | 0.1999 | | |
| Cor Total | 243.67 | 16 | | | |

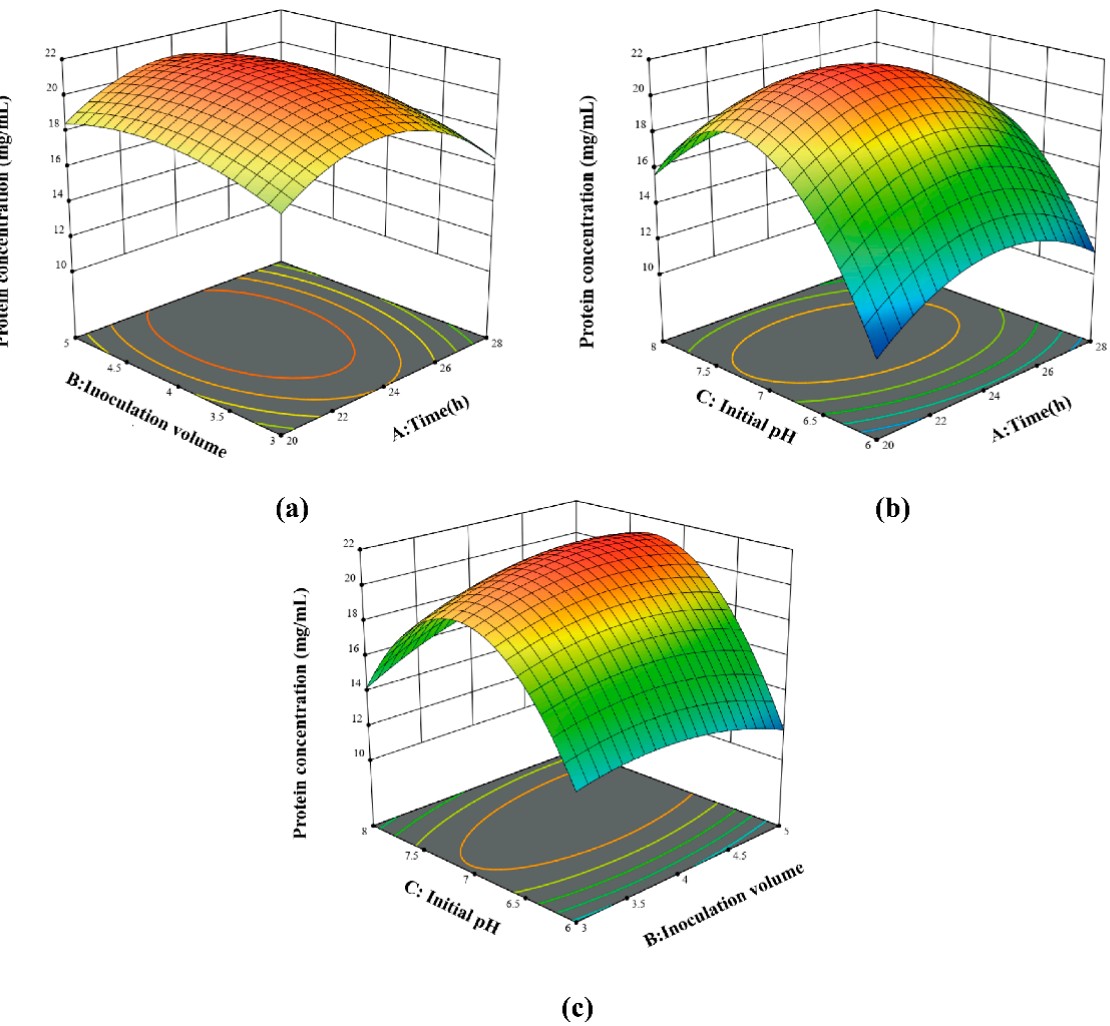

**Figure 3.** Effect of fermentation time and inoculum volume (**a**), initial pH and fermentation time (**b**), initial pH and inoculum volume (**c**) on protein concentration.

### 3.4. Analysis of Antioxidant Effects of Peptides with Different Molecular Weights

The antioxidant activity of peptides correlates with their hydrophobicity, molecular weight, and amino acid sequence [33]. Small peptides with low molecular mass have demonstrated higher antioxidant activity [30,34]. Zhang [12] and Hu [35] independently found that low-molecular-weight peptides exhibit high antioxidant activity during the extraction of wheat germ peptides and hickory protein peptides. The peptides undergo enzymatic cleavage by proteases produced by microorganisms, exposing them to more hydrophobic amino acids. This exposure enhances the action of peptides and unsaturated fatty acids, thereby facilitating more effective scavenging of free radicals [36,37]. The antioxidant activities of wheat germ peptides from different systems were determined. According to Figure 4a, vitamin C exhibits the highest DPPH free radical scavenging rate at 95.6%, followed by F5 at 88.5%, and F6 at 78.7%, which is marginally lower than the scavenging rate of F3 at 79.7%. As observed in Figure 4b, both F5 and F6 demonstrate significant scavenging ability against OH free radicals, with F5 achieving a clearance rate of 95.15% and F6 exhibiting scavenging ability comparable to that of the positive control. ABTS, being water-soluble free radicals, can reflect the overall antioxidant capacity of hydrophilic peptides [38]. Figure 4c illustrates that both F5 and F6 possess substantial ABTS free radical scavenging capacity, reaching 90.6% and 90.2%, respectively. The results indicate that both F5 and F6 exhibit considerable antioxidant capacity.

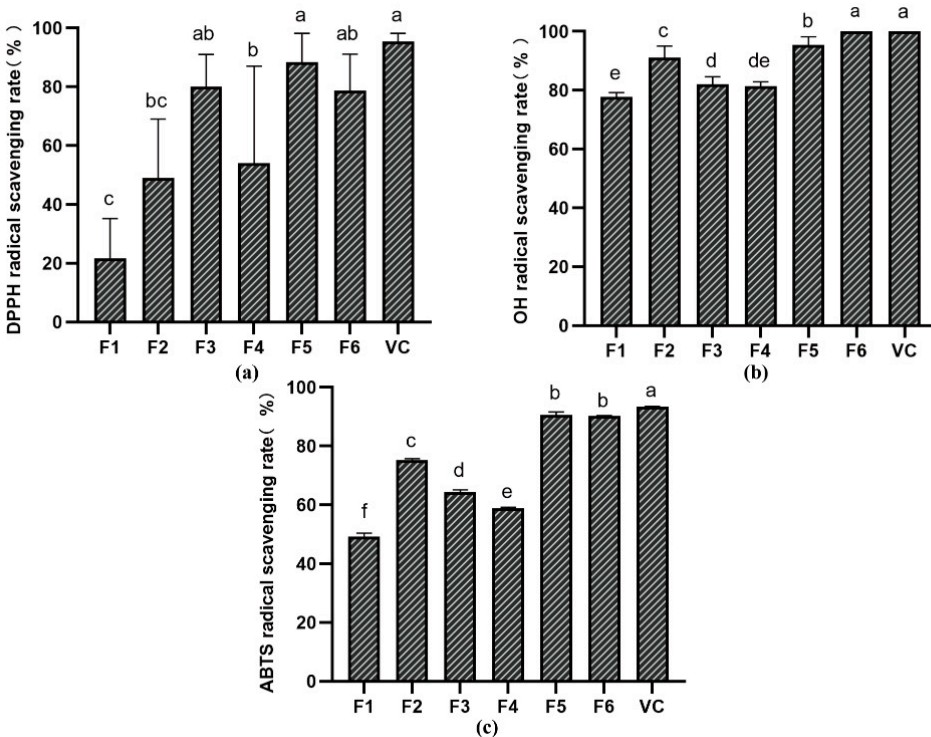

(F1: >100 kDa; F2: 50–100 kDa; F3: 30–50 kDa; F4: 10–30 kDa; F5: 3–10 kDa; F6: <3 kDa)

**Figure 4.** Analysis of antioxidant effects of peptides with different molecular weights: (**a**) DPPH free radical scavenging capacity; (**b**) OH free radical scavenging capacity; (**c**) ABTS free radical scavenging capacity (Different corresponding letters indicate significant differences at $p < 0.05$ by Duncan's test).

## 4. Discussion and Conclusions

We first utilized the reducing power for the screening of fermentation strains, and after fermentation by *Aspergillus niger*, the reducing power was increased from 0.41 to 1.08. This phenomenon may be attributed to the rich enzyme system of *Aspergillus niger*. Subsequently, *Aspergillus niger* was chosen to ferment the wheat germ protein solution. The optimal fermentation process for wheat germ protein peptides was determined using protein concentration as an indicator, resulting in a significant increase in protein concentration compared to pre-fermentation levels. Different molecular weight sizes of wheat germ peptides were obtained by using graded ultrafiltration to retain the fermented solution. The antioxidant activity of the isolated wheat germ peptides was assessed using 1,1-diphenyl-2-picryl-hydrazyl radical (DPPH), hydroxyl radical, and 2,2′-Azinobis-(3-ethylbenzthiazoline-6-sulphonate) (ABTS) scavenging assays. The findings revealed that wheat germ peptides with <3 KDa exhibited the highest overall antioxidant activity.

The enzymatic method of obtaining peptides, which is currently more available in the market, and the combination of this method and bioinformatics, which makes the whole process more controllable, make it a strategy with development potential. Additionally, the microbial fermentation method stands as an important approach for acquiring antioxidant peptides, attributed to its distinctive flavor and cost-effectiveness. The antioxidant capacity of these peptides is closely related to their amino acid composition, length, molecular structure, and hydrophobicity [39]. In addition, factors such as the type of substrate, the specificity of the enzyme, and the reaction time and temperature during the enzymatic hydrolysis may affect the molecular weight and amino acid sequence of the peptide, which in turn affect its biological activity [40]. *Aspergillus niger* is a common and important fermenting strain and one of the excellent producers of valuable proteases. The industrial products of *Aspergillus niger* fermentation currently primarily include citric acid, soy sauce, puerh tea, and other derivatives [41]. *Aspergillus niger* produces a diverse enzyme system, including acetyl esterases, amylases, glucosidases, mannanases, and others, contributing

significantly to the paper industry and the treatment of industrial fermentation wastewater [41,42]. The enzyme system produced by *Aspergillus niger* enzymatically cleaves wheat germ proteins to amino-acid-residue-exposed polypeptides, which are more favorable for binding to the moiety and acting activity. The antioxidant peptides extracted from traditionally semi-dried fermented fish in Thailand exhibit significant efficacy in suppressing free radicals and reactive oxygen species (ROS) [43]. This observed effect may be attributed to the presence of hydrophobic and aromatic amino acids within these peptides. The antioxidant activity of peptides is generally influenced by both the quantity of amino acids present and the specific sequence of the peptide. The active peptide derived from the proteolysis of wheat germ albumin has been demonstrated to possess antioxidant activity and a substantial content of free amino acids [44]. Peptides with molecular weight less than 3 kDa obtained by enzymatic hydrolysis of wheat embryo albumin have been shown to have higher free radical scavenging efficiency, and these peptides all contain highly bioactive amino acids [45]. After extracting the antioxidant peptide from shark skin, Yang ultimately determined, through chemical synthesis, that the interaction site between the peptide ATVY and ABTS free radicals was the N-terminal tyrosine [30]. Food-derived antioxidant peptides, as natural antioxidants, are of great value in promoting human health and disease intervention, and the preparation of antioxidant peptides from food protein sources, among others, has attracted much attention [46]. Nevertheless, further investigation is required to elucidate the mechanism of action of antioxidant peptides. Validation of functionality can be accomplished through cellular and animal experiments, coupled with comprehensive systematic studies involving proteomics, metabolomics, and other analyses of the mechanisms of action.

The optimal fermentation process parameters derived from the data analysis were a fermentation time of 24 h, inoculum amount of 4%, and initial pH = 7, and the predicted value of protein concentration after fermentation was 21.58 mg/mL. The validation test was carried out by using the above-mentioned process conditions, and the protein concentration was 21.47 mg/mL, which did not differ much from the theoretical predicted value. The reducing power of the fermented supernatant was higher than that of the unfermented supernatant; as shown in Table 4, the scavenging rates of DPPH, OH, and ABTS radicals by peptide F6 obtained after fermentation were increased by 43.66%, 25.47%, and 67.04%, respectively, compared with that of the fermented supernatant. Peptides with antioxidant activity were obtained by fermentation and ultrafiltration. This provides ideas for the preparation of antioxidant peptides from wheat germ.

**Table 4.** Comparison of antioxidant indexes of supernatant and wheat germ peptides after fermentation (F6: peptides with a molecular weight < 3 kDa).

| Clearance Rate | Fermentation Supernatant (%) | F6 (%) | Comparison (%) |
|---|---|---|---|
| DPPH | $54.79 \pm 0.13$ | $78.71 \pm 0.12$ | 43.66 |
| OH | $79.70 \pm 0.08$ | $100.00 \pm 0.01$ | 25.47 |
| ABTS | $54.00 \pm 0.01$ | $90.2 \pm 0.01$ | 67.04 |

**Author Contributions:** Conceptualization, J.H. and L.P.; methodology, Y.L. and Y.M.; investigation, J.W., C.Z. and Y.Z.; resources, X.C. and C.Z.; data curation, Y.L. and Y.H.; writing—original draft preparation, Y.L.; writing—review and editing, A.L., L.P. and X.C.; supervision, Y.H. and L.P.; project administration, J.H. and A.L.; funding acquisition, J.H. All authors have read and agreed to the published version of the manuscript.

**Funding:** This work was supported by the Cultivation Programme for Young Backbone Teachers in Henan University of Technology, High-Level Talents Research Fund of HAUT (Grant No. 2020BS064), Henan Province Science and Technology Research and Development Plan Joint Fund Project (222103810060), Henan Province Youth Science Fund Project (232300421266), and Key Research and Development Project of Henan Province (231111310700). This work was also financially supported by the Open Competition Research Projects of Xuchang University (2022JBGS11), Zhongyuan Scholars of Henan Province in China (192101510004), Zhongyuan Scholar Workstation Funded Project

(ZYGZZ2021056, 224400510026), and Central Government Guides the Local Science and Technology Development Special Fund (Z20221341069).

**Institutional Review Board Statement:** Not applicable.

**Informed Consent Statement:** Not applicable.

**Data Availability Statement:** The data that support the findings of this study are available from the corresponding author, upon reasonable request.

**Conflicts of Interest:** The authors declare no conflicts of interest.

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
