# Peer review of "Optimization of Fermentation Process of Wheat Germ Protein by Aspergillus niger and Analysis of Antioxidant Activity of Peptide"

_fermentation, doi:10.3390/fermentation10030121_

Round 1
Reviewer 1 Report
Comments and Suggestions for Authors
This was a very well described and presented research. The scientific level of the content was high and very appropriate. I suggest this article for publication. Some minor changes I propose to improve the quality of the paper prior to the publication are given below.
-Line 232: I would recommend you add the actual (uncoded) values of each factor/parameter of the design in Table 2 rather than the coded ones (i.e. -1, 0, 1).
-Lines 250-251: Is the polynomial equation in terms of uncoded units? If yes, please write it in terms of uncoded units (i.e. based on the actual factor values).
-Lines 291-331: I would suggest to combine the Discussion section (which is too short) with the Results. As I can see, you've already used literature to discuss your findings in the Results section, so this won't change much. According to the journal's guide for authors, there is available the option to combine the Discussion with the Results.
Comments on the Quality of English LanguageThe use of the English language is fine. However, the authors are advised to revise the text to improve some expressions and avoid any unintentional mistakes.
Reviewer 2 Report
Comments and Suggestions for Authors
Suggestions and corrections to the article can be found in the attached PDF.

Corrections to the English language of the article can be found in the attached PDF (fermentation-2807805-peer-review-v1.pdf)
Reviewer 3 Report
Comments and Suggestions for Authors
Totally speaking, this article is regarding the production, isolation, and purification of the biopeptides originated by the technological process of biotranformation (fermentation) products of the wheat germ. The current study's goal was to analyze the biopeptide acitivities after fermentation of wheat germ with the Aspergillus niger strain. Analysis of the antioxidant peptides and separation of them into a few peptide fractions by using the ultrafiltration technique was one of the research objectives. The manuscript was suitable for the Fermentation journal and the proposed Special Issue. However, there are some main points that require clarification.
(1) Affiliation section: Please include the matching e-mail addresses of the co-authors listed above, as well as their initials in parenthesis. Please, see the instructions for authors.
(2) Abstract section
There is a lack of a clearly written research goal, the reason for this kind of research, as well as the main objectives of the research. The methodology needs to be rewritten, specifically specifying which analysis was used to performed enzymatic hydrolysis of wheat germ protein via solid state or submerged fermentation process, which to determine the antioxidant activities of produces peptides. To say something about the ultrafiltration process (dead-and or cross-flow process) used, why it is used, where it is usually used, and what the purpose of its application is. Obtained experimental results nedostaju u kompletnom poglavlju. It is necessary to amend and supplement the Abstract section.
(3) Keywords section: It is necessary to add more key words in order to better adapt them to the manuscript of the scientific work. For instance: protease, biopeptides, fermentation, Aspergillus niger
(4) Introduction section
Please complete the introduction with specific statistical data on the utilization and production of whole wheat germ protein and bioactive peptides from wheat germ, as well as their production, isolation, and purification technological methods. The nutritional composition (especially protein content) of wheat germ must be listed in detail, including the citation of the necessary literature. Next, isolate the procedures for obtaining peptide fractions with improved bioactivities (i.e., a few illustrations of how the authors enhanced the characteristics of peptides using various endo- and egzo-peptodases).
(5) Materials and Methods section
Line 130: There is no information on the moisture content during the fermentation of wheat germ with Aspergillus niger. Also, there are no conditions under which fermentation was carried out, such as an incubator, shaker, or similar. So, write down the necessary information in the paper and clarify the method of fermentation.
Write how the supernatants are separated from the resulting biomass after the fermentation is completed. Write down how the protein content was determined and by which method. At the same time, the question arises: do all the proteins that are analyzed originate from the generated peptides, or do they also originate from the generated hydrolytic enzymes, products of Aspergillus niger metabolism? Have the activities of some enzymes, primarily proteases, been checked? If they are not, it is necessary to supplement the mentioned experiments, and in the discussion of the results, indicate what is necessary.
Line 148, “10w, 5w, 3w, 1w and 3000 Da”: The molecular masses that peptides can be separated into are not correctly written. Correct the error and write correctly. So, 10, 5, 3, 1, and 0.3 kDa.
Line 153: The antioxidant activity of bioactive peptides was analyzed through ABTS, DPPP, and OH radical tests. Since vitamin C was used as a positive control, it is necessary to express the results as μmol vitamin C equivalents/mg protein. Change both the methods and the display of the results.
The names of the devices (centrifuge and appropriated rotors and conditions; freeze-dryer, etc.) that were used in the whole experimental work and their manufacturers must be mentioned.
(6) Results and disscusion section
Subsection 3.1. Comparison of the reducing power of different strains of fermentation: Please, write the introductory part of the discussion of the results related to the reducing power and explain how the authors concluded that this particular A. niger strain was the most suitable for further research. All the strains that were analyzed produce large amounts of hydrolytic enzymes, which can further degrade complex substrates.
Line 198: The legend that will explain the abbreviations in the diagram is missing. Please, modify and add.
What about the fraction of polypeptides? Have they been analyzed? If not, it would be nice to know the fat content of the peptides in each of the extracted fractions.
Table 4: Explain in the manuscript how you calculated the comparison (%).
(7) Weather is conceivable for the authors to rephrase their conclusion section by drawing a general (concluded) influence on examined propeties and amino acid compositions i.e. molecular weight of isolated novel peptides?
(8) It is advised that the authors recheck the main text during the revision to make this manuscript more readable.
Reviewer 4 Report
Comments and Suggestions for Authors
I suggest a linguistic correction of the entire manuscript.
Round 2
Reviewer 2 Report
Comments and Suggestions for Authors
Follow the guidelines suggested below:
Lines 239 -248. The text should be before the Figure 1.
Lines 250, 255 and 260. Include The (article) before the sentences : Figure 2(a)........, Figure 2(b)...... and Figure 2(c)......
Line 321. ....,vitamin C.....
Lines 383-384. Add a legend about the term F6 in the Table 4.
Comments on the Quality of English LanguageFollow the guidelines suggested below:
Lines 239 -248. The text should be before the Figure 1.
Lines 250, 255 and 260. Include The (article) before the sentences : Figure 2(a)........, Figure 2(b)...... and Figure 2(c)......
Line 321. ....,vitamin C.....
Lines 383-384. Add a legend about the term F6 in the Table 4.
Author Response
Lines 239-248. Thank you for your suggestion. We have relocated the content to before Figure 1.
Lines 250, 255, and 260. Thank you for your suggestion. We have added the article "The" before the sentences.
Line 321. Thank you for your suggestion. We have corrected the spelling error here.
Lines 383-384. Thank you for your suggestion. We have annotated the term "F6" in the title.
Thank you for your valuable suggestions. We have incorporated your feedback and made the necessary revisions accordingly. Additionally, we have highlighted the changes in yellow for clarity.
We appreciate your thorough review and constructive input.
Reviewer 4 Report
Comments and Suggestions for Authors
The authors responded to the comments and made quite a few corrections to improve the quality of the article. The whole manuscript still needs to be read to correct minor errors (for example, in Line 66 Rhizobium should be in italics). Only the discussion, in my opinion, still lacks a proper positioning of the results against the background of the available literature.
Comments on the Quality of English LanguageOnly minor editing of English required
Author Response
Thank you very much for your suggestions. We have made the necessary changes by italicizing the name of the strain in line 66 and have also revised and expanded certain sections of the discussion. These modifications have been highlighted in yellow for your convenience in tracking the changes.
We trust that these adjustments and additions enhance the clarity and readability of the manuscript.
Once again, we appreciate your valuable feedback and thorough review. Should you have any further suggestions or modifications, we would be more than happy to accommodate them.